# Is there a role for neuregulin 4 in human nonalcoholic fatty liver disease?

**Toon J. I. De Munck**[1,2], **Markus Boesch**[3], **Pauline Verhaegh**[1,2], **Ad A. M. Masclee**[1,2], **Daisy Jonkers**[1,2], **Jos F. van Pelt**[4], **Johannie du Plessis**[3], **Hannelie Korf**[3], **Frederik Nevens**[3,5], **Ger H. Koek**[1,2], **Schalk Van der Merwe**[3,5], **Jef Verbeek**[3,5]*

1 Division of Gastroenterology and Hepatology, Department of Internal Medicine, Maastricht University Medical Centre, Maastricht, The Netherlands, 2 School of Nutrition and Translational Research in Metabolism (NUTRIM), Maastricht University, Maastricht, The Netherlands, 3 Laboratory of Hepatology, Department Chronic Diseases, Metabolism & Ageing (CHROMETA), KU Leuven, Leuven, Belgium, 4 Laboratory of Clinical Digestive Oncology, Department of Oncology, KU Leuven & University Hospitals Leuven and Leuven Cancer Institute (LKI), Leuven, Belgium, 5 Department of Gastroenterology and Hepatology, University Hospitals Leuven, Leuven, Belgium

* jef.verbeek@uzleuven.be

**Data Availability Statement:** Yes - all data are fully available without restriction.

**Funding:** J. Verbeek - seeding grant of the School of Nutrition and Translational Research in

## Abstract

### Background

Neuregulin 4 (Nrg4), a novel adipokine enriched in brown adipose tissue has been observed to negatively regulate de novo hepatic lipogenesis and limit nonalcoholic fatty liver disease (NAFLD) progression to nonalcoholic steatohepatitis (NASH) in rodents. However, the role of Nrg4 in human NAFLD remains unclear to date. We analysed Nrg4 plasma levels and its association with liver disease severity together with the transcriptional profile of the Nrg4 pathway in liver and visceral adipose tissue (VAT) of NAFLD patients.

### Methods

Plasma Nrg4 levels were measured in 65 NAFLD patients and 43 healthy controls (HC). Hepatic steatosis and fibrosis were diagnosed and quantified with chemical shift MRI and transient elastography respectively. Furthermore, blood lipid levels, HOMA-IR and systemic pro-inflammatory cytokines (TNF-α, IL-6 and IFN-γ) were analysed. Microarray analyses to assess differences in the Nrg4 and its receptor family ErbB pathway in liver and VAT from an independent patient group with biopsy proven NAFL (simple steatosis) (n = 4), NASH (n = 5) and normal liver (n = 6) were performed.

### Results

Plasma Nrg4 levels were not significantly different between NAFLD patients and HC (p = 0.622). Furthermore, plasma Nrg4 levels did not correlate with the hepatic fat fraction (r = -0.028, p = 0.829) and were not significantly different between NAFLD patients with or without hepatic fibrosis (p = 0.087). Finally, the expression profile of 82 genes related to the Nrg4-ErbB pathway in liver and VAT was not significantly different between NAFL, NASH or obese controls.

Metabolism (NUTRIM), Maastricht University, The Netherlands. https://www.maastrichtuniversity.nl/research/school-nutrition-and-translational-research-metabolism The funders had no role in study design, data collection and analysis, decision to publish, or preparation of the manuscript.

**Competing interests:** The authors have declared that no competing interests exist.

## Conclusion

Our study does not support a role for Nrg4 in the pathophysiology of human NAFLD.

## Introduction

Nonalcoholic fatty liver disease (NAFLD) affects 25% of the general population worldwide [1]. It is a disease spectrum ranging from simple steatosis (NAFL) to nonalcoholic steatohepatitis (NASH) that can lead to cirrhosis [2]. NAFLD is strongly associated with metabolic comorbidities including obesity, metabolic syndrome and type 2 diabetes. Recently, metabolic associated fatty liver disease (MAFLD) has been proposed as a new name for NAFLD, reflecting its complex pathophysiology in a milieu of systemic metabolic dysfunction [3, 4]. Multiple factors such as adipose tissue dysfunction, insulin resistance, dietary habits, microbiome perturbations and genetic predisposition all contribute to the development and progression of NAFLD [5]. Recently the role of brown adipose tissue (BAT) in the pathophysiology of NAFLD has gained interest after the discovery of BAT in adults and its reduced activity in overweight and obese subjects [6]. An inverse correlation between BAT presence (measured with PET-CT) and NAFLD prevalence was observed in humans [7]. BAT can increase energy expenditure through heat production (via the unique expression of uncoupling protein 1) and is therefore a potential therapeutic target for metabolic diseases like obesity, type 2 diabetes and NAFLD [8]. In addition, BAT secretes regulatory molecules—also called "batokines"—such as neuregulin 4 (Nrg4), fibroblast growth factor 21 (FGF-21), interleukin-6 (IL-6) and insulin-like growth factor 1 (IGF1) [9] that might directly target peripheral tissues such as liver, white adipose tissue (WAT), pancreas, and bone in an endocrine manner [9].

Three pre-clinical studies in mice suggested a protective role of Nrg4 for the development and progression of NAFLD [10–12]. Nrg4 is a member of the epidermal growth factor (EGF) family of extracellular ligands and is highly expressed in adipose tissue, mainly in BAT and to a lesser extend in WAT. Wang *et al.* observed that Nrg4 deficiency exacerbates hepatic steatosis and insulin resistance in mice [10]. They showed that Nrg4 activates ErbB3/ErbB4 signalling in cultured primary mouse hepatocytes and negatively regulates de novo lipogenesis mediated by the LXR/SREBP1c pathway. Furthermore, Guo *et al.* observed an increase in the degree of hepatic fibrosis and hepatic inflammation associated with Nrg4 deficiency in a NASH mouse model suggesting that Nrg4 plays a role in the progression from simple steatosis to NASH [11]. Finally, Ma *et al.* observed that, in mice, gene transfer mediated Nrg4 overexpression blocks high-fat diet induced weight gain, insulin resistance and hepatic lipogenesis [12].

However, only two studies assessed the potential role of Nrg4 in human subjects with NAFLD [13, 14]. One study observed decreased Nrg4 plasma levels in children with NAFLD compared to children without NAFLD [13]. Another study observed decreased plasma levels of Nrg4 in adults with NAFLD compared with those without, but no association with the degree of hepatic steatosis quantified with ultrasound was observed [14].

Thus despite promising data in animal studies, the pathophysiological significance of Nrg4 in human NAFLD remains unclear to date. Therefore, in this study, we assessed the level of circulating Nrg4 in our well-characterized cohort of NAFLD patients and compared these with healthy controls. Furthermore, the association of plasma Nrg4 with the degree of hepatic steatosis (measured by chemical shift MRI) and systemic inflammatory and metabolic parameters were assessed. Finally, in an independent cohort, via microarray analysis we assessed

hepatic and visceral adipose tissue gene expression changes in the Nrg4 pathway in relation to biopsy-assessed NAFLD severity.

## Methods

### Study populations and definition of metabolic variables

All patients were included between July 2015 and April 2019 in our Maastricht NAFLD cohort. Participants were recruited via the outpatient clinics of the division of Hepatology of the Maastricht University Medical Centre+ and Zuyderland medical hospital and via CO-EUR, a second line treatment facility for eating disorders. Inclusion criteria were (1) age between 18 and 65 years, (2) NAFLD diagnosis based on imaging (any form of imaging showing steatosis) or biopsy and confirmation of steatosis by MRI fat quantification and (3) (BMI) $\geq$25 kg/m$^2$. Exclusion criteria were excessive alcohol use (>7 units/week for women, >14 units/week for men), secondary causes of hepatic fat accumulation (e.g. alcohol, medication, Wilson's disease, viral infections, starvation or parenteral nutrition), pregnancy or breastfeeding, diagnosis of liver cirrhosis and current or prior diagnosis of malignancy within the last 5 years. Anthropometric measurements, venous blood sample, transient elastography (fibroScan$^\circledR$) and chemical shift MRI of the liver were performed in each participant. Healthy controls (HC) were recruited via public advertising on websites and in local newspapers. HC were aged between 18 and 67 years with a BMI < 25 kg/m$^2$, had no excessive alcohol consumption ($\leq$7 units/week for women, $\leq$14 units/week for men) and had identical exclusion criteria compared to the NAFLD subjects. This study was approved by the Medical Ethics Committee of the Maastricht University Medical Centre (ClinicalTrials.gov Identifier: NCT02422238) and performed according the declaration of Helsinki (latest amendment of 2013, Fortaleza, Brazil). Prior to participation, all participants gave written informed consent.

Liver biopsy procedure and histological assessment were performed as previously described [15]. Insulin resistance (IR) was defined by means of the homeostasis model of assessment-insulin resistance (HOMA-IR) with the following formula HOMA-IR = (fasting plasma insulin ($\mu$IU/mL) x fasting plasma glucose (mmol/L))/22.5 [16]. In subjects on insulin therapy (n = 3), HOMA-IR was not calculated. Metabolic syndrome was diagnosed based on the updated International Diabetes Federation (IDF) definition with abdominal obesity defined as waist circumference $\geq$94cm or $\geq$80 cm in male and female participants respectively [17]. The definition of type 2 diabetes mellitus (DM2) at baseline included any patient with an HbA1c $\geq$6.5% or a fasting plasma glucose of $\geq$7 mmol/l or a patient with current treatment for DM2 [18].

In an independent cohort of 15 obese patients, the expression of Nrg4 related genes in liver and visceral adipose tissue were assessed via microarray analysis. These 15 patients were selected based on the availability of both liver and fat biopsies from our previously published study in 113 patients from three academic hospitals (Leuven, Antwerp and Pretoria). These patients underwent liver biopsy and collection of visceral adipose tissue (VAT) during bariatric surgery [15]. A detailed description of the inclusion and exclusion criteria of these subjects is reported previously [15]. In brief, subjects were included between 2010 and 2011. Liver and VAT were obtained during bariatric surgery and liver histology was assessed by an expert liver pathologist blinded to all clinical information according to the NASH–Clinical Research Network Scoring System criteria [19]. Patients with inadequate liver biopsy specimens (<2 cm and/or <5 portal tracts), with underlying cirrhosis, or borderline features of NASH (NAFLD activity score [NAS], 3–4) were excluded from the analysis. Patients were divided in three groups: (1) obese: less than 5% steatosis, NAFLD activity score (NAS) of 0 (n = 6); (2) NAFL: NAFLD without inflammation or NASH, NAS less than 4 (n = 4); (3) NASH: NAS of 5 or

greater and fibrosis score of 0–1 (n = 5). This study was approved by the the Medical ethics committee of KU Leuven/UZ Leuven, the ethical committee of the Antwerp University Hospital and the University of Pretoria Ethics Committee. Prior to participation, all participants gave written informed consent.

## Plasma Nrg4 and cytokine analyses

Fasting venous blood samples were collected from each subject and stored at − 80˚C until analysis. When performing the assay, samples were brought to room temperature. Plasma Nrg4 levels were measured by using enzyme-linked immunosorbent assay (ELISA) (Catalogue No. CSB-EL016080HU; Cusabio, Wuhan, China) according to the guidelines of the manufacturer. Plasma levels of interleukin 6 (IL-6), tumor necrosis factor alpha (TNF-α), and interferon gamma (IFN-γ) were assessed in duplicate on multiplex V-Plex kits from Meso Scale Discovery (MSD, Meso Scale Discovery, Rockville, USA) according to the guidelines of the manufacturer.

## Magnetic resonance hepatic fat quantification

Upon inclusion, all NAFLD patients underwent a chemical shift MRI (1,5T or 3T Philips) of the liver to diagnose NAFLD and quantify the hepatic fat fraction (HFF). A commercially available version of mDIXON sequence package was used to acquire fat, water, in-phase (IP) and opposed-phase (OP) images. HFF was measured on three MRI sections. On each section four circular regions of interest (ROIs) of 5 cm$^2$ were drawn in the liver. Artefact, vascular and biliary structures were avoided and the ROI was copied from the IP image OP image. The mean signal intensity (SI) loss of all 12 ROIs was calculated with the following formula: (SI IP-SI OP)/ 2*SI IP*100% [20]. Based on a previous study comparing this method with MR spectroscopy the applied cut-off to diagnose hepatic steatosis was 3.6% [21].

## Transient elastography

In all NAFLD patients liver stiffness measurement (LSM) was performed using the FibroScan® (touch 502, Echosens, Paris, France). Both the M probe (3,5 MHz) and the XL probe (2,5 MHz) were available for this study. Only measurements with at least 10 valid measurements (during one examination), a minimal success rate of 60% and interquartile range/median for liver stiffness estimation ≤ 30% were accepted for further analysis. Examinations that did not meet these conditions were classified as failures. The final result of LSM is the median of the valid (at least 10) individual measurements. The applied FibroScan® cut-off value for histological grade 2 fibrosis (F2, significant fibrosis) or grade 3 fibrosis (F3, advanced fibrosis) were 7.0 kPa and 8.7 kPa, respectively [22].

## Microarray analysis

Microarray analysis was based on the analysis of the previous microarray readings [15]. In short, microarray hybridization was performed according to the manufactures instructions on Affymetrix Primeview arrays (Affymetrix, Inc, Santa Clara, CA). Additionally, pathway analysis related to Nrg4 and its receptor ErbB was performed with the internet-based DAVID Bioinformatics Resource 6.8 program (National Institute of Allergy and Infectious Diseases [NIAID], NIH, Bethesda, MD). We identified the Kyoto Encyclopedia of Genes and Genomes (KEGG) Has:04012 ErbB signaling pathway with 85 genes that were further investigated and compared in the microarray results of visceral adipose tissue (VAT) and liver. Three genes were excluded, as they were not hybridized and if gene duplicates were

found, the one with the higher Robust Multi-array Average (RMA) expression were kept. The results were analyzed in the R programming environment version 4.0.2 and visualized with Pheatmap version 1.0.12.

## Statistical analysis

Data were analysed with IBM SPSS statistics version 25 (IBM Statistics for Macintosh, Chicago, IL, USA). The normality of data distribution was evaluated visually and by means of shapiro-wilk test. Data are presented as means ± SD for normally distributed continuous variables and as percentages for categorical variables, or as median (interquartile range; 25th percentile to 75th percentile) for continuous variables lacking a normal distribution. Student's test was used to compare normal distrusted data between two groups, while Mann-Whitney U-tests was used to compare skewed distributed variables between two groups. Chi-square test was used to compare categorical variables between groups. As demographic factors may influence the relationship between NAFLD and circulating Nrg4 levels a multivariable linear regression analysis was performed. Spearman's rho correlation coefficient was used to analyse the correlation between plasma Nrg4 levels and the hepatic fat fraction, blood lipid profile, markers of insulin resistance and inflammatory cytokines levels. Microarray data were compared between groups using a moderated t-statistic [23, 24] implemented in the limma package (version 3.12.1) and correcting for multiple testing with Benjamini-Hochberg [25]. A two-sided P-value of <0.05 was considered statistically significant. Missing data was not imputed, but reported upon in the results. We did not perform a sample size calculation as this study was performed as a sub-analysis of a longitudinal cohort study. Nrg4 was analysed in all included participants of this cohort study until April 2019.

## Results

### Baseline characteristics

Plasma Nrg4 levels were measured in 65 NAFLD patients and 43 healthy controls (HC). Demographic, anthropometric and biochemical parameters of all participants are presented in Table 1. NAFLD patients were older and had a higher BMI than HC. Furthermore, plasma levels of TNF-α and IL-6 were significantly higher in NAFLD patients compared to HC (p<0.001). Twenty-two NAFLD patients (38%) had clinically significant fibrosis (transient elastography value of ≥7 kPa).

### Plasma Nrg4 levels in NAFLD patients with and without DM2 compared to healthy controls

Plasma Nrg4 levels were not significantly different between NAFLD patients and HC (respectively 89.2 [68.6; 102.2] vs. (89.5 [68.2; 112.2]; data expressed as median and [interquartile range (Q1, Q3)] (p = 0.622) (Fig 1A). Previous reports showed that the presence of type 2 diabetes may influence the level of circulating NRG4 levels [26]. We observed no difference in plasma Nrg4 levels between NAFLD patients with DM2 (n = 16), NAFLD patients without DM2 (n = 49) and HC (n = 43) (P = 0.381) (Fig 1B). As gender and age may influence Nrg4 levels and thus their relationship with the presence of NAFLD, a multivariable linear regression was performed. The multivariable models include age, gender and presence of NAFLD (yes vs. no) as independent variables. In this model no association between the presence

**Table 1. Baseline characteristics.**

| Variable (standard value) | NAFLD (n = 65) | HC (n = 43) | P-value |
|---|---|---|---|
| Gender (female) | 34 (52%) | 22 (51%) | 0.724 |
| Age (years) | 54 (45;61) | 40 (25; 52) | 0.001 |
| BMI (kg/m2) | 32.37 (±4.75) | 22.24 (±2.10) | <0.001 |
| Metabolic Syndrome (Yes) | 39 (61%) | 0 (0%) | <0.001 |
| DM2 (Yes) | 16 (25%) | 0 (0%) | <0.001 |
| HbA1c (4.4–6.2%) | 5.6 (5.3; 6.5) | - | |
| Fasting Glucose (3.1–6.1mmol/L) | 6.0 (5.3; 6.8) | - | |
| HOMA-IR (n = 58) | 3.29 (2.32; 5.11) | - | |
| Total cholesterol (5.0–6.4 mmol/L) | 5.11 (±0.95) | - | |
| LDL-C (3.5–4.4 mmol/L) | 3.13 (±0.93) | - | |
| HDL-C (>0.9 mmol/l) | 1.26 (±0.41) | - | |
| Triglycerides (0.80–1,94 mmol/l) | 1.49 (1.05; 2.29) | - | |
| ALT (F <34, M <45U/L) | 40 (29; 60) | - | |
| AST (F <31, M <35U/L) | 28 (24; 38) | - | |
| γ-GTP (F <38, M <55U/L) | 42 (25; 67) | - | |
| AP (F <98, M <115U/L) | 90 (76; 106) | - | |
| Ferritin (F 20–150, M 20–250 ug/l) | 132 (61; 238) | - | |
| Albumin (35–55 g/l) | 40.68 (±3.14) | - | |
| CRP (<6 mg/l) | 2 (1; 4) | - | |
| Hepatic fat fraction (< 5%) | 16.86 (11.86; 26.14) | - | |
| LSM (kPa) (n = 60) | 6.1 (5.1–8.1) | - | |
| • F2 (7–8.7 kPa) (yes/total) | 8/60 (13%) | | |
| • ≥ F3 (≥8.7 kPa) (yes/total) | 14/60 (25%) | | |
| TNF-α (pg/ml) | 3.02 (2.34; 3.73) | 2.14 (1.70; 2.60) | <0.001 |
| IL-6 (pg/ml) | 0.50 (0.34; 0.67) | 0.23 (0.14; 0.37) | <0.001 |
| IFN-γ (pg/ml) | 3.15 (2.26; 5.33) | 3.38(1.84; 5.05) | 0.566 |

Values are presented as mean (±SD) (normal distributed data) or median (Q1; Q3) (skewed data) for continuous variables or number (% of group) for categorical variables. Student's T-test (normal distributed data) or Mann-Whitney U tests (skewed data) was used to determine statistically significant differences (P < 0.05) between groups. Chi-square test was used to compare categorical variables between groups. HOMA-IR was not available in seven subjects, three used insulin therapy and in four subjects data were missing. In five participants FibroScan® results were not obtained. F, Female; M, Male; BMI, body mass index; HbA1c, hemoglobin A1c; LDL-C, low-density lipoprotein cholesterol; HDL-C, high-density lipoprotein cholesterol; ALT, alanine transaminase; AST, aspartate transaminase; γ-GTP, gamma-glutamyltranspeptidase; AP, alkaline phosphatase; LSM, liver stiffness measurement; TNF- α, tumor necrosis factor alpha; IL-6, interleukin 6; IFN-γ, interferon gamma.

NAFLD and circulating Nrg4 levels was observed (B = 7.773 (95% CI -1.891; 17.357)) in NAFLD patients (for detail see S1 Table).

## Association between plasma Nrg4 levels and hepatic steatosis

No correlation between the hepatic fat fraction measured by MRI and plasma Nrg4 levels was observed in NAFLD patients (r = -0.028, p = 0.829) (Fig 2). No statistically significant difference in plasma Nrg4 levels between NAFLD patients with fibrosis (F2-3, ≥7.0 kPa) and NAFLD patients without fibrosis (F0-2, <7.0 kPa) was observed (F0-1: 84.5 [63.2; 97.9], F2-3: 95.8 [69.6; 112.9] (median [Q1, Q3]) P = 0.087).

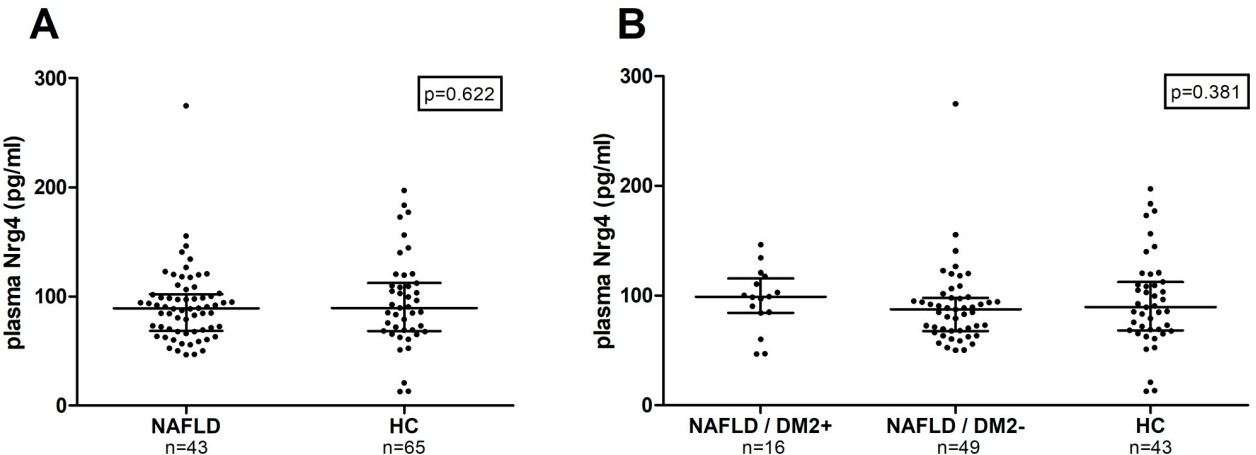

**Fig 1. Plasma Nrg4 levels in NAFLD patients versus healthy controls.** Values are presented in scatter plots with median line (Q1, Q3). Plasma Nrg4 levels were compared between groups with the Mann-Whitney U-tests (A) and Kruskall Wallis test (B). Plasma Nrg4 levels in (A) HC versus NAFLD patients, (B) NAFLD patients with and without DM2 and HC. DM2, type 2 diabetes mellitus.

## Correlation of plasma Nrg4 levels with metabolic risk factors in NAFLD patients

No correlation of plasma Nrg4 levels with BMI, fasting glucose, HbA1c, HOMA-IR, cholesterol and triglycerides levels was observed (Table 2). Plasma Nrg4 levels only were moderately negatively correlated with HDL-Cholesterol ($\rho$ = -0.254, P = 0.041). Pro-inflammatory cytokines might reduce the expression of Nrg4 [10]. However, we found no correlation between Nrg4 levels and plasma pro-inflammatory cytokines. All investigated correlations were visualised in scatter plots in S1 Fig. Also in the complete study population (including NAFLD patients and HC) no correlation between plasma Nrg4 and BMI, TNF-α, IL-6 and IFN-γ (n = 108) was found (all p> 0.05; S2 Table).

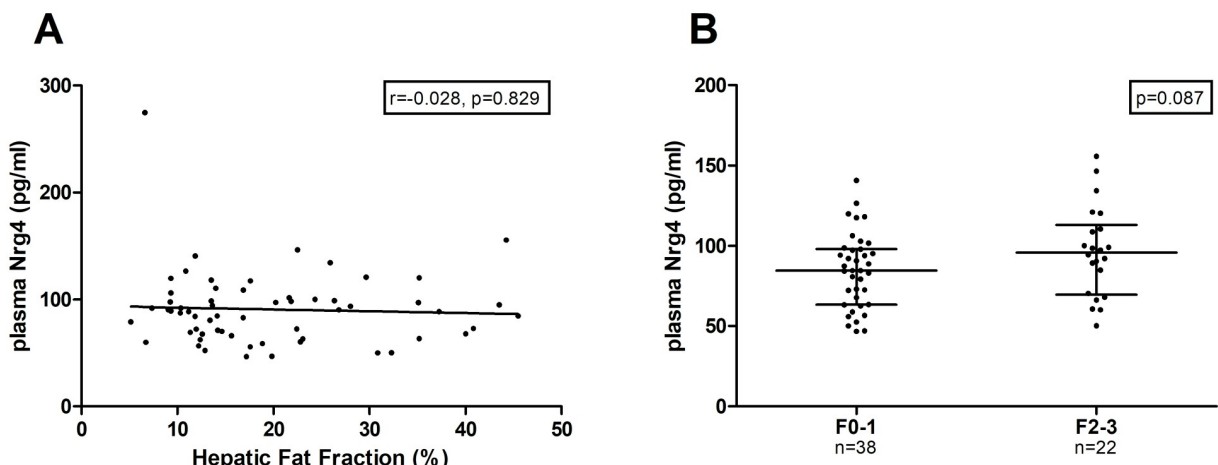

**Fig 2. Association between plasma Nrg4 levels and NAFLD severity.** Values are presented in scatter plots with median line and (Q1, Q3). Spearman correlation was used for correlation analysis and Mann-Whitney U-tests for comparison between two groups. (A) Spearman correlation between hepatic fat fraction and plasma Nrg4 levels in NAFLD patients. (B) Plasma Nrg4 levels in NAFLD patients without clinically significant hepatic fibrosis (F0-1), versus NAFLD patients with clinically significant hepatic fibrosis (F2-3).

**Table 2. Spearman's rank correlations with plasma Nrg4 levels and metabolic parameters in NAFLD patients (n = 65).**

|  | Spearman's rank (ρ) | p-value |
|---|---|---|
| **BMI** | 0.005 | 0.967 |
| **Fasting Glucose** | 0.196 | 0.126 |
| **HbA1c** | 0.221 | 0.077 |
| **HOMA-IR (n = 58)** | 0.127 | 0.341 |
| **Total Cholesterol** | -0.164 | 0.192 |
| **HDL-C** | -0.254 | 0.041 |
| **LDL-C** | -0.116 | 0.357 |
| **Triglycerides** | 0.147 | 0.244 |
| **TNF-α** | -0.138 | 0.273 |
| **IL-6** | -0.185 | 0.140 |
| **IFN-γ** | 0.124 | 0.324 |

BMI, body mass index; HbA1c, haemoglobin A1c; LDL-C, low-density lipoprotein cholesterol; HDL-C, high-density lipoprotein cholesterol; TNF- α, tumor necrosis factor alpha; IL-6, interleukin 6; IFN-γ, interferon gamma.

## Microarray analysis on the NrG4-ErbB signalling pathway in the liver and visceral adipose tissue in relation to NAFLD severity

Baseline characteristics of 15 obese subjects included in the microarray analysis are presented in S3 Table. Hepatic and visceral adipose tissue gene expression profiles were compared between six obese subjects without NAFLD, four NAFL and five NASH patients (Fig 3). No significant difference was seen in Nrg4 expression between subjects without NAFLD, with NAFL or with NASH. Additionally, none of the average Robust Multi-array Average (RMA) expressions in the 82 investigated genes related to the Nrg4-ErbB signalling pathway in both liver and visceral adipose tissue were altered between these groups (Fig 3). This means the KEGG:04012 ErbB pathway was under no condition significantly enriched. The Robust Multi-array (RMA) expressions of the 82 investigated genes related to the Nrg4-ErbB signalling pathway of all individual subjects are presented in S1 Fig.

## Discussion

Nrg4 has been identified as an endocrine factor secreted by brown adipose tissue that selectively binds to the liver and inhibits hepatic lipogenesis in mice and hepatocyte cell culture [10]. Therefore, it is crucial to explore whether these findings can be translated to human NAFLD. In the present study, we did not find a difference of plasma Nrg4 levels between well-characterized NAFLD patients and HC. Furthermore, plasma Nrg4 levels were not correlated with the degree of hepatic steatosis measured by chemical shift MRI in NAFLD patients. In an independent cohort, microarray analysis revealed no hepatic gene expression changes related to NrG4 and its receptor family ErbB between biopsied obese patients with NAFL, NASH and normal livers.

In this study, we provide the first data on Nrg4 in a European NAFLD cohort. Plasma Nrg4 levels were only studied in two Asian NAFLD studies (one pediatric, one adult) that both observed decreased plasma Nrg4 levels in NAFLD patients compared to controls [13, 14]. This discrepancy might be attributed to difference in study population and design. In both studies ultrasound was used to diagnose and quantify hepatic steatosis. However, ultrasonography has poor sensitivity to detect and quantify hepatic steatosis and is therefore suboptimal to investigate the correlation between circulating Nrg4 levels and the presence or degree of hepatic

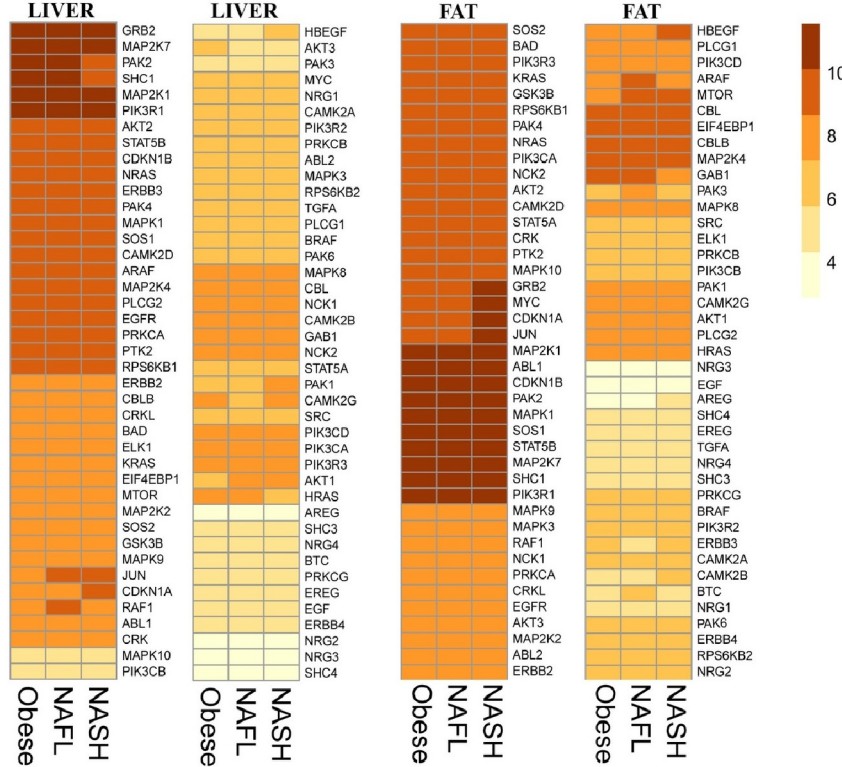

**Fig 3. Heatmap of 82 NrG4-ErbB signalling pathway related genes.** Visualised as average Robust Multi-array Average (RMA) expressions in the liver and visceral adipose tissue of obese, NAFLD and NASH patients.

steatosis [27]. Chemical shift MRI on the other hand is an accurate and reproducible method to diagnose and quantify hepatic steatosis [28]. Differences in BAT volume and activity have been observed between different ethnicities and should be subject of further study [29].

The liver is the target tissue of Nrg4 in mice, likely through its direct binding to the ErbB receptors [10]. Therefore, we investigated the expression of 82 genes of the Nrg4 related ErbB signalling pathway in the liver of obese patients without NAFLD, NAFL and NASH patients without fibrosis. No significant difference in enrichment of any of the genes was observed, arguing against a significant role of Nrg4 in NAFLD pathophysiology. Furthermore, it has been observed that Nrg4 mRNA levels in human adipose tissue inversely correlated with body fat mass and liver fat content [10]. However, our microarray analysis did not show a difference of Nrg4 expression level in visceral adipose tissue between the same NAFL, NASH and controls. This discrepancy might be related to a difference in patient number. Wang *et al.* performed mRNA analysis in 642 adipose tissue samples, although the method of hepatic fat quantification and the number of subjects in which liver fat was quantified was not mentioned [10]. Nrg4 binding is restricted to the liver in rodents, but it remains to be shown that Nrg4 signalling is restricted to the liver in humans as well given the wide-spread expression of ErbB3 and ErbB4 in humans [30]. Our microarray analysis did not reveal significant changes of any of the ErbB pathway related genes in VAT between the different groups. This suggests that if plasma Nrg4 may act on VAT tissue, this effect is not altered in NAFLD patients.

To elucidate the mechanism behind the adipose tissue Nrg4 deficiency in obese mice, treatment of brown and white adipocytes with TNF-α was performed and resulted in a decreased Nrg4 expression in BAT and WAT [10]. Therefore, it was hypothesised that the systemic

pro-inflammatory activity associated with the metabolic syndrome contributes to the obesity-induced downregulation of Nrg4 in BAT and WAT [10]. To investigate this hypothesis, we analysed plasma TNF-α, IL-6 and IFN-γ levels in our study population. Although plasma TNF-α and IL-6 levels were significantly increased in NAFLD patients compared to HC, none of the pro-inflammatory cytokines correlated with plasma Nrg4 levels. Further studies are needed to explore the mechanism behind Nrg4 induction and secretion.

The overall discrepancy between the animal studies and our data could be partly explained by the difference in Nrg4 analysis. While we measured plasma Nrg4, Wang *et al.* solely focused on *Nrg4* expression in tissues [10]. Circulating Nrg4 levels are the result of Nrg4 secretion not only by adipocytes, but also by liver, pancreas and muscles [31]. Plasma Nrg4 levels therefore not necessarily perfectly correlate with mRNA expression in an individual organ or tissue. However, in addition to our assessment of plasma Nrg4, also via microarray analysis we found no changes in the Nrg4/ErbB pathway in liver and visceral adipose tissue.

In previous studies, NAFLD-related metabolic factors including BMI, HOMA-IR or DM2 were both positively [32–34] and negatively [13, 35–37] associated with circulating Nrg4 levels in humans. In line with a recent meta-analysis on circulating Nrg4 levels in diabetic patients, we observed no correlation between BMI, HOMA-IR, fasting glucose, blood triglycerides levels and plasma Nrg4 levels [26]. The clinical significance of the observed negative but weak correlation between plasma Nrg4 levels and HDL-C, in line with the finding of Chen *et al.* in diabetic individuals, is doubtful given the unclear role of BAT function on cholesterol levels [38] and even might be the result of multiple testing error [33, 34, 37].

Our study has some limitations. Firstly, the current gold standard to diagnose and stage NAFLD, a liver biopsy, was not performed in our NAFLD cohort in which we analysed Nrg4 plasma levels. However, chemical shift MRI is a highly valid and reproducible alternative to qualitatively and quantitatively assess hepatic fat content [39]. Furthermore, compared to liver biopsy, MRI hepatic fat quantification is not prone to sampling error. Secondly, causality between circulating Nrg4 levels and hepatic steatosis or fibrosis development could not be investigated because of the cross-sectional nature of the study. Thirdly, HC were not screened for the presence of hepatic steatosis but had no history of liver disease, diabetes or hypertension and a BMI <25 kg/m$^2$ and therefore NAFLD prevalence is expected to be below 7% [40]. Finally, HC were significantly younger than NAFLD patients. However, in a recent study age was not correlated with circulating Nrg4 levels in 311 DM2 patients [37].

In conclusion, no difference in plasma Nrg4 levels between NAFLD patients and HC were observed and plasma Nrg4 levels were not correlated with the degree of hepatic steatosis, insulin sensitivity and systemic pro-inflammatory cytokines. Finally, gene expression profiles of the Nrg4 related ErbB pathway in liver and VAT were not significantly changed between NAFL, NASH or obese controls. Therefore, our study does not support the pre-clinical finding that Nrg4 deficiency associates with increased hepatic lipogenesis and NAFLD progression. Notwithstanding, future human studies are needed to further assess the pathophysiological role and therapeutic potential of Nrg4 in human NAFLD.

## Supporting information

**S1 Table. Multivariable regression analyses of plasma Nrg4 levels in the total study population (n = 108).** B: unstandardized regression coefficient, 95% CI: 95% confidence interval. The multivariable models include age, gender and presence of NAFLD as independent variables. Assumptions for linear regression were met since there were no influential outliers based on Cooks distance ≤ 0.184, and collinearity was met as indicated by variance inflation factor

values ≤ 1.201. R square of the model was 0.054.
(DOCX)

**S2 Table. Spearman's rank correlations between plasma Nrg4 levels and plasma inflammatory markers in the total study population (n = 108).** TNF- α, tumor necrosis factor alpha; IL-6, Interleukin 6; IFN-γ, interferon gamma.
(DOCX)

**S3 Table. Baseline characteristics of the patients used for microarray analysis of liver and visceral fat tissue.** Values are presented as median (Q1; Q3) (skewed data) for continuous variables or number (% of group) for categorical variables. LDL-C, low-density lipoprotein cholesterol; HDL-C, high-density lipoprotein cholesterol; ALT, alanine transaminase; AST, aspartate transaminase; γ-GTP, gamma-glutamyltranspeptidase; AP, alkaline phosphatase.
(DOCX)

**S1 Fig. Scatter plots of plasma Nrg4 levels and metabolic parameters in NAFLD patients (n = 65).** Scatter plot of Nrg4 levels with (A) BMI, (B) Fasting glucose, (C) HbA1C, (D) HOMA-IR, (E) total cholesterol, (F) HDL-C, (G) HDL-C, (H) Triglycerides, (I) TNF-α, (J) IL-6 and (K) IFN-γ. BMI, body mass index; HbA1c, haemoglobin A1c; LDL-C, low-density lipoprotein cholesterol; HDL-C, high-density lipoprotein cholesterol; TNF-α, tumor necrosis factor alpha; IL-6, interleukin 6; IFN-γ, interferon gamma.
(PDF)

**S2 Fig. Heatmap of 82 NrG4-ErbB signalling pathway related genes in all individual participants.** Robust Multi-array (RMA) expressions in the liver and visceral adipose tissue of obese, NAFLD and NASH patients.
(PDF)

## Author Contributions

**Conceptualization:** Jef Verbeek.

**Data curation:** Pauline Verhaegh, Ger H. Koek.

**Formal analysis:** Toon J. I. De Munck, Markus Boesch, Jos F. van Pelt.

**Funding acquisition:** Jef Verbeek.

**Investigation:** Toon J. I. De Munck, Markus Boesch.

**Methodology:** Jef Verbeek.

**Resources:** Ad A. M. Masclee, Daisy Jonkers, Schalk Van der Merwe.

**Supervision:** Jef Verbeek.

**Writing – original draft:** Toon J. I. De Munck.

**Writing – review & editing:** Pauline Verhaegh, Ad A. M. Masclee, Daisy Jonkers, Jos F. van Pelt, Johannie du Plessis, Hannelie Korf, Frederik Nevens, Ger H. Koek, Schalk Van der Merwe, Jef Verbeek.

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
