## [Decision Letter · Decision Letter 0]

5 Mar 2021

PONE-D-21-01793

Is there a role for neuregulin 4 in human nonalcoholic fatty liver disease?

PLOS ONE

Dear Dr. Verbeek,

Thank you for submitting your manuscript to PLOS ONE. After careful consideration, we feel that it has merit but does not fully meet PLOS ONE’s publication criteria as it currently stands. Therefore, we invite you to submit a revised version of the manuscript that addresses the points raised during the review process.

The manuscript was reviewed by two experts in the field and there was consensus that minor revision was required prior to publication. 

We look forward to receiving your revised manuscript.

Kind regards,

Robin D Clugston, Ph.D.

Academic Editor

PLOS ONE

Journal Requirements:

3)  Please provide a sample size and power calculation in the Methods, or discuss the reasons for not performing one before study initiation.

4)  In your Methods section, please provide additional information about the participant recruitment method and the demographic details of the independent cohort of 15 participants. Please ensure you have provided sufficient details to replicate the analyses such as: a) the recruitment date range (month and year), b) a description of any inclusion/exclusion criteria that were applied to participant recruitment, c) a description of how participants were recruited.

5) Thank you for including your ethics statement: "This study was approved by the Medical Ethics Committee of the Maastricht University Medical Centre approved the protocol (ClinicalTrials.gov Identifier: NCT02422238) and performed according the declaration of Helsinki (latest amendment of 2013, Fortaleza, Brazil). Prior to participation, all participants gave written informed consent."   

a. Please amend your current ethics statement to include the full name of the ethics committee/institutional review board(s) that approved the study from which the independent cohort of 15 obese patients were taken.

b. Please state whether written informed consent was obtained from the independent cohort of 15 obese patients.

6) PLOS requires an ORCID iD for the corresponding author in Editorial Manager on papers submitted after December 6th, 2016. Please ensure that you have an ORCID iD and that it is validated in Editorial Manager. To do this, go to ‘Update my Information’ (in the upper left-hand corner of the main menu), and click on the Fetch/Validate link next to the ORCID field. This will take you to the ORCID site and allow you to create a new iD or authenticate a pre-existing iD in Editorial Manager. Please see the following video for instructions on linking an ORCID iD to your Editorial Manager account: https://www.youtube.com/watch?v=_xcclfuvtxQ

7) Please include captions for your Supporting Information files at the end of your manuscript, and update any in-text citations to match accordingly. Please see our Supporting Information guidelines for more information: http://journals.plos.org/plosone/s/supporting-information.

8) We note that you have included the phrase “data not shown” in your manuscript. Unfortunately, this does not meet our data sharing requirements. PLOS does not permit references to inaccessible data. We require that authors provide all relevant data within the paper, Supporting Information files, or in an acceptable, public repository. Please add a citation to support this phrase or upload the data that corresponds with these findings to a stable repository (such as Figshare or Dryad) and provide and URLs, DOIs, or accession numbers that may be used to access these data. Or, if the data are not a core part of the research being presented in your study, we ask that you remove the phrase that refers to these data.

Reviewers' comments:

Reviewer's Responses to Questions

**Comments to the Author**

1. Is the manuscript technically sound, and do the data support the conclusions?

Reviewer #1: Yes

Reviewer #2: Yes

2. Has the statistical analysis been performed appropriately and rigorously? 

Reviewer #1: Yes

Reviewer #2: Yes

3. Have the authors made all data underlying the findings in their manuscript fully available?

Reviewer #1: No

Reviewer #2: Yes

4. Is the manuscript presented in an intelligible fashion and written in standard English?

Reviewer #1: Yes

Reviewer #2: Yes

5. Review Comments to the Author

Reviewer #1: This is an interesting well-conducted study. I have only a few specific points, all addressable with current data

Specific points:

The authors should provide further clarification of the microarray studies. They refer to their previous Gastroenterology patient, in which indeed, 15 patients had both adipose tissue and liver biopsies. I understand that these 15 patients were selected on the basis of availability of liver samples. If this was not the criteria, how were these 15 patients selected?

It is unclear from the report what is the starting point for the microarray analysis: previous frozen samples, previously isolated and preserved RNA, or was based on the analysis of the previous microarray readings (that have been made available in a public repository with the previous paper). If it is the latter, in the case of fat analysis (in which there were samples from 35 patients) is there any reason why the authors did not analyze all patients?

In the report of the microarrays, especially taking into account the small number of samples, the authors should provide additional heatmaps showing expression of individual patients (it is ok to show the means in main paper)

The authors point out that age was different between controls and NAFLD. The authors could show in a regression analysis that age does not explain the lack of differences in NRG4 levels between healthy controls and NAFLD

Since fibroscan provides a quantitative assessment of fibrosis (more or less imprecise), the association between NRG4 levels and fibrosis should be showed with the continues values of fibroscan, rather than categorizing the fibroscan values. Fig 2B should be like figure 2A, showing the scatterplot fibroscan vs plasma NRG4

The metrics of correlations shown in table 2 are not very informative for the reader. The authors should provide all the scatterplots as supplementary data.

Reviewer #2: The subject investigated by De Munck et al. in their manuscript is of interest. Despite evidence in mouse models that Nrg4 (likely to be originated in brown fat and perhaps in other adipose depots) may exert hepatoprotective effects for NAFLD, the relevance of these findings in humans is unclear. De Munck et al. show an associational study of NRG4 levels variations (plasma levels, gene expression) in patients with distinct conditions in relation to NAFLD. Despite the results were negative (in the sense that no clear correlations are found between NRG4/NRG4-mediated pathways and signs of pathology) data are worth to be reported in light of existing doubts on the role of NRG4 in human liver pathophysiology.

The study is reasonably performed and, despite the associational character of the study in relation to establishment of case-and-effect evidence (a limitation intrinsic to most human studies in this area), conclusions appear sound.

Suggestions for improvement are:

- In table 1, provision of standard values of metabolites, glucose/insulin parameters, hepatic enzymes, etc in the HC group (page 9) would had improved quality in the presentation of the cohort/s.

- I recommend that the authors perform a correlation study of TNFa and IL6 levels in relation to the NALFD-related parameters,..it would be interesting to see whether these inflammation-related parameters associate with NAFLD or parameters of metabolic disease in their cohort.

- Some statements provided in a clear-cut manner in Discussion: " microarray analysis did not show a difference of Nrg4 expression levels,..." (lines 293-4) or "none of the pro-inflammatroy cytokines correlated with plasma Nrg4 levels" (lines 309-310) are difficult to be identified as such in the Results section. Please include them explicitly in the Results section for clarity.

- Consider the possibility to validate at least a few key gene expression data (e.g. NRG4 gene expression itself) from microarray analysis using a qRT-PCR specific assay.

6. PLOS authors have the option to publish the peer review history of their article (what does this mean?). If published, this will include your full peer review and any attached files.

Reviewer #1: No

Reviewer #2: No

---

## [Author Response · Author response to Decision Letter 0]

24 Apr 2021

SEE REBUTTAL LETTER

Journal Requirements 

1. Review Reference list 

Checked. 

2. Change style to PLOS ONE’s requirements 

Checked. 

3. Please provide a sample size and power calculation in the Methods, or discuss the reasons for not performing one before study initiation.

We did not perform a sample size calculation as this study was performed as a sub-analysis of our longitudinal Maastricht NAFLD cohort study. Nrg4 was analysed in all included participants of this cohort study until April 2019. Additionally, the number of analysed patients is in line with other studies assessing the role of plasma Nrg4 (1-2). Now, we added to the Method section following statement: We did not perform a sample size calculation as this study was performed as a sub-analysis of our longitudinal Maastricht NAFLD cohort study. Nrg4 was analysed in all included participants of this cohort study until April 2019.

1. Wang R, Yang F, Qing L, Huang R, Liu Q, Li X. Decreased serum neuregulin 4 levels associated with non-alcoholic fatty liver disease in children with obesity. Clin Obes. 2019 Feb;9(1):e12289. doi: 10.1111/cob.12289. Epub 2018 Nov 8. PMID: 30411515.

2. Kang YE, Kim JM, Choung S, Joung KH, Lee JH, Kim HJ, Ku BJ. Comparison of serum Neuregulin 4 (Nrg4) levels in adults with newly diagnosed type 2 diabetes mellitus and controls without diabetes. Diabetes Res Clin Pract. 2016 Jul;117:1-3. doi: 10.1016/j.diabres.2016.04.007. Epub 2016 Apr 23. PMID: 27329015.

4. In your Methods section, please provide additional information about the participant recruitment method and the demographic details of the independent cohort of 15 participants. Please ensure you have provided sufficient details to replicate the analyses such as: a) the recruitment date range (month and year), b) a description of any inclusion/exclusion criteria that were applied to participant recruitment, c) a description of how participants were recruited.

A detailed description of the inclusion and exclusion criteria of these subjects is reported in our previous manuscript (du Plessis J, et al. Association of Adipose Tissue Inflammation With Histologic Severity of Nonalcoholic Fatty Liver Disease. Gastroenterology. 2015;149(3):635-48.e14.). In brief, subjects were included between 2010 and 2011. Liver and VAT was obtained during bariatric surgery and liver histology was assessed by expert liver pathologists blinded to all clinical information according to the NASH–Clinical Research Network Scoring System criteria. Patients with inadequate liver biopsy specimens (<2 cm and/or <5 portal tracts), with underlying cirrhosis, or borderline features of NASH (NAFLD activity score [NAS], 3–4) were excluded from the analysis. This was added to the method section of the manuscript. 

5. Please amend your current ethics statement to include the full name of the ethics committee/institutional review board(s) that approved the study from which the independent cohort of 15 obese patients were taken. Please state whether written informed consent was obtained from the independent cohort of 15 obese patients.

This study was approved by the Medical ethics committee (KU leuven/UZ leuven), the ethical committee of Antwerp University Hospital and the University of Pretoria Ethics Committee. Prior to participation, all participants gave written informed consent. This was added to the method section of the manuscript. 

6. Please include captions for your Supporting Information files at the end of your manuscript, and update any in-text citations to match accordingly.

Updated. 

7. We note that you have included the phrase “data not shown” in your manuscript….. 

Now, these data are added in full in the supplementary material section. 

Reviewer 1

1. The authors should provide further clarification of the microarray studies. They refer to their previous Gastroenterology patient, in which indeed, 15 patients had both adipose tissue and liver biopsies. I understand that these 15 patients were selected on the basis of availability of liver samples. If this was not the criteria, how were these 15 patients selected? 

These 15 patients were indeed selected based on the availability of both liver and fat biopsies from our previously published study in 113 patients from three academic hospitals (Leuven, Antwerp and Pretoria), who underwent liver biopsy and collection of visceral and subcutaneous fat tissue during bariatric surgery. In other words, liver in combination with fat tissue was indeed only available in 15/113 subjects. This was added to the method section of the manuscript. 

2. It is unclear from the report what is the starting point for the microarray analysis: previous frozen samples, previously isolated and preserved RNA, or was based on the analysis of the previous microarray readings (that have been made available in a public repository with the previous paper). If it is the latter, in the case of fat analysis (in which there were samples from 35 patients) is there any reason why the authors did not analyze all patients?

The current analysis of the Nrg4 pathway was based on the analysis of the previous microarray readings from our/Prof van der Merwes lab that is online available (Transcript profiling: accession numbers: GSE58979 and GSE59045). Indeed, we do have more patients in which the fat was analyzed, however we wanted to use the same patients for combined liver and fat analysis. We clarified this in the method section of the manuscript. 

3. In the report of the microarrays, especially taking into account the small number of samples, the authors should provide additional heatmaps showing expression of individual patients (it is ok to show the means in main paper). 

We thank the reviewer for this comment. We added a supplementary heatmap (2S Fig.) as you suggested. Now the expression of individual patients is also presented. 

4. Since fibroscan provides a quantitative assessment of fibrosis (more or less imprecise), the association between NRG4 levels and fibrosis should be showed with the continues values of fibroscan, rather than categorizing the fibroscan values. Fig 2B should be like figure 2A, showing the scatterplot fibroscan vs plasma NRG4

We respectfully chose not to analyze the fibroscan results as continuous values. Probably, there is no clinical relevance (i.e. difference in quantity of fibrosis) in the difference between for example 3 kPa and 6 kPa Next to this limited discriminative ability for low-fibrosis stages, the real added diagnostic value of fibroscan lies in the exclusion of advanced liver disease/fibrosis. Therefore we chose to present the fibroscan results in a categorized manner. In order to address the remark of the reviewer, we also checked if there was any correlation between continuous fibroscan results and Nrg4 levels, which was not the case (pearson correlation = 0.202, p = 0.122). 

5. The metrics of correlations shown in table 2 are not very informative for the reader. The authors should provide all the scatterplots as supplementary data.

As suggested, we provided all scatter plots in a supplementary figure (S1 Fig.)

Reviewer 2

1. In table 1, provision of standard values of metabolites, glucose/insulin parameters, hepatic enzymes, etc in the HC group (page 9) would had improved quality in the presentation of the cohort/s.

We agree, therefore standard values were added in table 1. 

2. I recommend that the authors perform a correlation study of TNFa and IL6 levels in relation to the NALFD-related parameters,..it would be interesting to see whether these inflammation-related parameters associate with NAFLD or parameters of metabolic disease in their cohort.

We thank the reviewer for the suggestion. However, the main objective of this study was to investigate the association of circulating Nrg4 levels with hepatic and metabolic variables in NAFLD patients. In addition, we already published a manuscript on the association between these pro-inflammatory markers and NAFLD disease severity (du Plessis J, et al. (2016) Pro-Inflammatory Cytokines but Not Endotoxin-Related Parameters Associate with Disease Severity in Patients with NAFLD. PLoS ONE 11(12): e0166048. https://doi.org/10.1371/journal.pone.0166048). Therefore, we decided not to perform these analyses again in the current manuscript. 

3. Some statements provided in a clear-cut manner in Discussion: " microarray analysis did not show a difference of Nrg4 expression levels,..." (lines 293-4) or "none of the pro-inflammatroy cytokines correlated with plasma Nrg4 levels" (lines 309-310) are difficult to be identified as such in the Results section. Please include them explicitly in the Results section for clarity.

We agree with the reviewer. Therefore, we adapted the results section : 

No significant difference was seen in Nrg4 expression between subjects without NAFLD, with NAFL or with NASH. Additionally, none of the average Robust Multi-array Average (RMA) expressions in the 82 investigated genes related to the Nrg4-ErbB signalling pathway in both liver and visceral adipose tissue were altered between these groups (Fig. 3). In addition, S2 fig was added to make this more clear for the reader. 

4. Consider the possibility to validate at least a few key gene expression data (e.g. NRG4 gene expression itself) from microarray analysis using a qRT-PCR specific assay.

We agree this would be interesting, but unfortunately we do not have enough samples left of the included patients. However, previously, we confirmed other micro-array results with qRT-PCR (e.g. CCL2, CCL21, CCL3, GADD45B and more) in these patients (ref: du Plessis J, et al. Association of Adipose Tissue Inflammation With Histologic Severity of Nonalcoholic Fatty Liver Disease. Gastroenterology. 2015;149(3):635-48.e14.). Therefore, we feel confident the same applies to Nrg4. Furthermore, microarray is a robust technique we applied in a sufficient number of patients (N=4-6). Taken together, we feel confident that our results indicate there is no change in NRG4 in the context of NAFLD.

---

## [Decision Letter · Decision Letter 1]

4 May 2021

Is there a role for neuregulin 4 in human nonalcoholic fatty liver disease?

PONE-D-21-01793R1

Dear Dr. Verbeek,

We’re pleased to inform you that your manuscript has been judged scientifically suitable for publication and will be formally accepted for publication once it meets all outstanding technical requirements.

Kind regards,

Robin D Clugston, Ph.D.

Academic Editor

PLOS ONE

Additional Editor Comments (optional):

Reviewers' comments:

Reviewer's Responses to Questions

**Comments to the Author**

1. If the authors have adequately addressed your comments raised in a previous round of review and you feel that this manuscript is now acceptable for publication, you may indicate that here to bypass the “Comments to the Author” section, enter your conflict of interest statement in the “Confidential to Editor” section, and submit your "Accept" recommendation.

Reviewer #1: All comments have been addressed

2. Is the manuscript technically sound, and do the data support the conclusions?

Reviewer #1: (No Response)

3. Has the statistical analysis been performed appropriately and rigorously? 

Reviewer #1: (No Response)

4. Have the authors made all data underlying the findings in their manuscript fully available?

Reviewer #1: (No Response)

5. Is the manuscript presented in an intelligible fashion and written in standard English?

Reviewer #1: (No Response)

6. Review Comments to the Author

Reviewer #1: (No Response)

7. PLOS authors have the option to publish the peer review history of their article (what does this mean?). If published, this will include your full peer review and any attached files.

Reviewer #1: No

---

## [Editor Report · Acceptance letter]

6 May 2021

PONE-D-21-01793R1 

Is there a role for neuregulin 4 in human nonalcoholic fatty liver disease? 

Dear Dr. Verbeek:

I'm pleased to inform you that your manuscript has been deemed suitable for publication in PLOS ONE. Congratulations! Your manuscript is now with our production department. 

Kind regards, 

on behalf of

Dr. Robin D Clugston 

Academic Editor

PLOS ONE